# Halogenated Pyrrolopyrimidines with Low MIC on *Staphylococcus aureus* and Synergistic Effects with an Antimicrobial Peptide

**DOI:** 10.3390/antibiotics11080984

**Published:** 2022-07-22

**Authors:** Cecilie Elisabeth Olsen, Fredrik Heen Blindheim, Caroline Krogh Søgaard, Lisa Marie Røst, Amanda Holstad Singleton, Olaug Elisabeth Torheim Bergum, Per Bruheim, Marit Otterlei, Eirik Sundby, Bård Helge Hoff

**Affiliations:** 1Department of Chemistry, Norwegian University of Science and Technology (NTNU), NO-7491 Trondheim, Norway; cecilie.e.olsen@ntnu.no (C.E.O.); fredrik.h.blindheim@ntnu.no (F.H.B.); 2Department of Clinical and Molecular Medicine, Norwegian University of Science and Technology (NTNU), NO-7489 Trondheim, Norway; caroline.d.sogaard@ntnu.no (C.K.S.); amanda.singleton@ntnu.no (A.H.S.); olaug.e.t.bergum@ntnu.no (O.E.T.B.); marit.otterlei@ntnu.no (M.O.); 3Department of Biotechnology and Food Science, Norwegian University of Science and Technology (NTNU), NO-7491 Trondheim, Norway; lisa.m.rost@ntnu.no (L.M.R.); per.bruheim@ntnu.no (P.B.); 4Department of Material Science, Norwegian University of Science and Technology (NTNU), NO-7491 Trondheim, Norway; eirik.sundby@ntnu.no

**Keywords:** antibacterial, pyrrolopyrimidines, synergistic effect, halogenated antibiotic

## Abstract

Currently, there is a world-wide rise in antibiotic resistance causing burdens to individuals and public healthcare systems. At the same time drug development is lagging behind. Therefore, finding new ways of treating bacterial infections either by identifying new agents or combinations of drugs is of utmost importance. Additionally, if combination therapy is based on agents with different modes of action, resistance is less likely to develop. The synthesis of 21 fused pyrimidines and a structure-activity relationship study identified two 6-aryl-7*H*-pyrrolo [2,3-*d*] pyrimidin-4-amines with potent activity towards *Staphylococcus aureus*. The MIC-value was found to be highly dependent on a bromo or iodo substitution in the 4-benzylamine group and a hydroxyl in the *meta* or *para* position of the 6-aryl unit. The most active bromo and iodo derivatives had MIC of 8 mg/L. Interestingly, the most potent compounds experienced a four-fold lower MIC-value when they were combined with the antimicrobial peptide betatide giving MIC of 1–2 mg/L. The front runner bromo derivative also has a low activity towards 50 human kinases, including thymidylate monophosphate kinase, a putative antibacterial target.

## 1. Introduction

Due to the rise in antimicrobial resistance, there is an urgent need of novel antibiotics preferably acting on new biological targets. Moreover, if therapy is based on two or more agents with differing modes of action, resistance is less likely to develop. Several studies have shown a synergistic effect between antimicrobial peptides (AMP) and traditional antibiotics, both in lowering the minimum inhibitory concentration (MIC) and overcoming resistance [1,2,3,4,5,6,7,8]. Synergistic effects can happen by several mechanisms, such as increased uptake [5] or though changes in metabolic nucleotide demand which possibly makes bacteria more sensitive to nucleotide anti-metabolites [6]. We, therefore, wanted to explore this avenue and selected betatide, a cell-penetrating peptide that targets the bacterial DNA sliding clamp [9], which has previously demonstrated substantial MIC lowering when it is combined with classical antibiotics [10].

As a starting point to discover new low molecular weight antibiotics, we selected PKI-166 (Figure 1), a pyrrolopyrimidine that was previously investigated as an anti-cancer agent [11] and recently identified as a thymidylate monophosphate kinase inhibitor (TMPK) [12]. Fused pyrimidines in general constitute versatile scaffolds for medicinal chemistry and are present in several FDA-approved drugs [13,14]. They have a bioisosteric relationship to purines and can, therefore, bind to a broad range of proteins [15,16]. Experimental pyrrolopyrimidine antibiotics often inhibit groups of similar bacterial targets [17,18], although some have relatively narrow profiles, such as the inhibitors towards bacterial DNA topoisomerases GyrB and ParE that were developed by Tari et al. [19,20] and the inhibitors of thymidylate monophosphate kinase (TMPK) that was published by Blindheim et al. [12]. TMPK is a key enzyme in the synthesis of thymidine diphosphate, an essential nucleoside in DNA synthesis [21]. This enzyme is located at the junction between the de novo and salvage pathways of thymidine triphosphate synthesis [22], and its inhibition could, therefore, be bactericidal by invoking the thymineless death response [23]. Thymineless death is a phenomenon where bacteria that are starved of thymine nucleosides experience detrimental events, including single- and double-strand DNA-breaks, plasmid elimination, and a loss of transforming ability [23].

Finally, we wanted to explore the effect of halogen insertion on antibacterial activity. These elements are rarely found in drugs besides X-ray contrast agents [24] and the only heavy halogen-containing drug on the WHO 2021 AWaRe list [25] was brodimoprim [26], which is not in current clinical use [27]. One reason that brominated and iodinated drugs are rare could be that hit selection metrics disfavor these as starting points in drug development campaigns. However, heavier halogen-containing antibiotics have been described from many different origins such as natural products [28,29], by microbial cultivation with halogen salts, [30], and synthetic origin [31,32,33,34]. Brominated tryptophanes and tyrosines are also frequently used in peptides with antibiotic effects [35]. Increased activity and stability of heavy halogen-modified peptide antibiotics has also been reported [36]. This effect could be due to the altered size and concomitant change in van der Waals interactions, but could also be caused by directional interactions with the target through halogen bonding [37,38].

Herein we describe a structure-activity relationship study which shows that increased activity and selectivity can be achieved by the insertion of heavy halogens (bromine and iodine). Further, the most active agents in combination with betatide showed a clear synergetic effect towards *S. aureus*. In the search of antibacterial targets, an enzymatic assay towards *E. coli* and human TMPK was performed. Additionally, the most active derivative was evaluated toward a panel of 50 human kinases to identify off-targets.

## 2. Results

### 2.1. Study Design

Our previous study attempting to identify *E. coli* TMPK inhibitors revealed that the previously discontinued epidermal growth factor receptor (EGFR) inhibitor PKI-166 [10,12] had an IC_50_ of 15 µM against *E. coli* TMPK (Figure 1). We, therefore, wanted to develop this lead further by aiming to (i) reduce the EGFR inhibitory activity, (ii) improve antibacterial activity, and possibly (iii) develop a better structure-activity relationship (SAR) understanding.

From our previous EGFR studies, we knew that the inhibitory activity of this kinase was very sensitive to substitution of the amine part [10]. Therefore, we first investigated variations of the 4-amino group, which led to the discovery that *p*-bromo substitution (compound **5**) had improved antibacterial activity with a concomitant reduction in EGFR activity. Then, further SAR investigation was performed by modifying the 6-aryl group and exchanging the NH in the pyrrolopyrimidine with a sulphur giving the corresponding thienopyrimidine.

In contrast to our previous studies on TMPK inhibitors [10,12] where enzymatic assays were employed, herein we have chosen to assay the compounds in culture to ensure that the cell penetrating compounds are identified. *E. coli* was selected as a model of Gram-negative bacteria and *S. aureus* for Gram-positive bacteria. Finally, we planned to investigate the synergetic effect when combining a small molecular antibiotic with antimicrobial peptides.

### 2.2. Synthesis of Fused Pyrimidines

The synthesis in the pyrrolopyrimidine series started from 4-chloro-7*H*-pyrrolo [2,3–*d*]pyrimidine, which was protected with *N*-(trimethyl) silylethoxymethyl (SEM) and selectively iodinated in the 6-position. This yielded building block **22** with two “handles” that could easily be modified in Pd-catalyzed cross-coupling reactions or nucleophilic aromatic substitutions. Compound **22** was then used to construct all pyrrolopyrimidine derivatives.

In the first route (Figure 2), we started with a Suzuki coupling on **22** with 4-hydroxyphenyl boronic acid as coupling partner and Pd_2_(dba)_3_ as a catalyst. This led to a high selectivity for the mono 6-arylated **23** over the 4,6-diarylated byproduct (19:1 ratio). Yields in the range of 52–71% were obtained. In most cases the aminations at C-4 proceeded smoothly but required significant amounts of co-base when the amine HCl salts were used. Acid catalyzed reactions were unsuccessful. The removal of the SEM-protecting group was then achieved by a two-step protocol using trifluoroacetic acid (TFA) and NaHCO_3_ in a previously established procedure [12]. This led to a library of pyrrolopyrimidines in 13-98% yields. Some debenzylation occurred for the more electron-rich amines, and an overall high crystallinity complicated chromatographic purification. We found purification by reversed phase chromatography to work better than normal phase in most cases. An alternative variant of this route was also evaluated by utilizing the methoxy analog **24**. However, purification after Suzuki coupling was found to be more difficult due to coelution, and the double deprotection at the end both involved the rather harsh reagent boron tribromide and was challenging to monitor by HPLC and ^1^H NMR spectroscopy.

To study the antibacterial effect of varying the 6-aryl substitution pattern, we used the route that is shown in Figure 3. Here, amination of **22** with the most active amine ((*R*)--1-(4-bromophenyl)ethan-1-amine) gave the advanced building block **34** in 91% yield, see Figure 3. In the following chemoselective Suzuki cross-coupling, six different palladium catalysts were evaluated, of which Pd(dppf)Cl_2_ was found to be the most selective. Depending on the nature of the substitution and scale, the selectivity varied between 3:1 and 19:1 in favor of the mono-arylated vs. the diarylated by-product and yields in the range of 26–72% were obtained. Finally, SEM deprotection afforded a series of substituted 6-aryl pyrrolopyrimidines in 36–78% yield. The aniline derivative **15** was made by reduction of the nitro analogue **14**.

Finally, the effect of substituting the pyrrole nitrogen was investigated by the construction of two thienopyrimidine derivatives. Here, 6-bromo-4-chloro-thienopyrimidine was coupled with 4-methoxyphenyl boronic acid to yield building block **44**, which was aminated with (*R*)-1-(4-bromophenyl)ethan-1-amine to give thienopyrimidine **20**, containing a *para*-methoxy group, see Figure 4. Demethylation with BBr_3_ yielded the corresponding phenolic derivative **21**.

### 2.3. Evaluation of Antibiotic Activity

#### 2.3.1. Variation of the 4-Amino Group

Our previous efforts at targeting TMPK using enzyme assays have, unfortunately, arrived at compounds that are inactive in culture experiments [12,39]. In this study, we have, therefore, altered our strategy and used minimal inhibitory concentration assays to identify and improve antibacterial agents. A total of 10 pyrrolopyrimidines with variation of the 4-amino group were assayed for their MIC in *E. coli* (MG1655) and *S. aureus* (ATCC29213) using the broth microdilution method [40]. Subsequently, the compounds were also counter-screened for their inhibition of human EGFR. The results are summarized in Table 1.

Given its activity towards *E. coli* TMPK in enzymatic studies [12], it was unfortunate that **1** and all the other derivatives did not possess any activity towards *E. coli* in culture. This is probably due to the more complex cell wall of Gram-negative bacteria and, therefore, lack of cellular uptake. On the other hand, compound **1** had an MIC of 64 mg/L towards the Gram-positive *S. aureus*. Whereas the corresponding 4-fluoro derivative **3** was inactive, increased potency was seen for the other *para*-substituted derivatives and an MIC of 8 mg/L towards *S. aureus* was seen for the *para*-bromo and *para*-iodo derivatives **5** and **6**. The MIC was increased to 16 mg/L with a *meta*-bromine substituent, which might indicate a steric clash. Interestingly, the removal of the methyl group at the stereocenter (comp. **8**), changing the stereochemistry (comp. **9**) or adding an *N*-methyl (comp. **10**) abolished all activity towards *S. aureus*. The relatively large variation in the MIC values upon these minor structural changes strongly suggests that the compounds act on an intracellular target, rather than having some unspecific effect on the bacterial cell membrane. Finally, the enzymatic assay towards human EGFR showed that the activity dropped when the size of the 4-aryl substituent increased, in line with our earlier assumption.

#### 2.3.2. Variation of the 6-Aryl Group and the 7-Heteroatom

In the second series of potential inhibitors, we varied the 6-aryl group and the heteroatom of the five-membered heterocycle. The results are summarized in Table 2. The pyrrolopyrimidines **11**–**18** were all inactive towards both *E. coli* and *S. aureus*, but the *meta*-hydroxy derivative **19** had an MIC of 16 mg/L towards *S. aureus*. Further, whereas the methoxy substituted thienopyrimidine **20** had no activity, the corresponding *para*-hydroxy derivative **21** had an MIC of 32 mg/L. This clearly shows the important role of both the pyrrole NH and the *para*- or *meta*-hydroxy group for the activity of this compound class.

#### 2.3.3. Combination Studies with the Antimicrobial Peptide Betatide

The derivatives with an MIC < 64 mg/L were then subjected to additional MIC profiling towards *S. aureus* in combination with the antimicrobial peptide betatide, which has demonstrated antimicrobial effects in multiple multi-resistant bacterial strains [9,10]. The concentration of betatide was 8 mg/L, which corresponds to half the MIC concentration.

The results that are summarized in Table 3 showed that all the compounds experience increased activity and lower MIC when they are combined with betatide. The activity trend roughly followed that which was seen for the single agents with the *para*-bromo and the *para*-iodo derivatives **5** and **6** being most potent.

#### 2.3.4. Structure-Activity Relationship

The SAR information that was gathered is shown in Figure 1. Crucial for activity is a phenolic group in *para* or *meta* position of the 6-aryl group. The other substitution patterns that were tested resulted in no activity.

Although some activity is retained when the heteroatom in the five membered ring is substituted for sulphur, the corresponding pyrrolopyrimidines were more active. For the 4-amino group, a single methyl group in R_3_ switches activity on and off. Thus, R_2_ should be a methyl and R_3_ must be a hydrogen. Other alkyls as R_2_ were not tested. For the R_1_-substituents at the benzylamine there was a preference for *para*-substitution by bromine and iodine. Further studies on the variation of the R_1_ group must be performed to verify if the increase in the activity is caused by the halogens or if it is purely a size effect.

### 2.4. Kinase Off-Targets and Mechanistic Studies

Based on the similarity of **5** with structures in our previous study [12], one could assume the bromo derivative **5** to be a TMPK inhibitor. Unfortunately, an assay towards *S. aureus* TMPK was not available. Instead, compound **5** was assayed towards the human and *E. coli* variant of TMPK, see Figure 2. Compound **5** had an IC_50_ of 5 µM towards *E. coli* TMPK and >100 µM towards the human variant. Thus, even though the sequence similarity is not high (34% by Protein Blast), TMPK could be a target also in *S. aureus* as the folding (see Appendix A) and important residues in the catalytic domains: LID, *p*-loop, and the DRX motif are highly conserved [41,42].

As the starting point of this study was the EGFR inhibitor PKI-166 (**1**), there is an obvious risk of kinase off-target effects for the bromo derivative **5**. Therefore, we profiled **5** towards a panel of 50 human kinases. The most inhibited kinase was EGFR (91% inhibition at 500 nM test concentration), while very low inhibition was seen towards all the other kinases that were tested. This is shown in Figure 3. A follow up IC_50_ measurement towards EGFR showed PKI-166 (**1**) to have IC_50_ < 1 nM, while **5** had an IC_50_ of 25 nM. Based on our previous experience [43,44], it is assumed that the EGFR inhibiting activity of **5** is too low to cause in vivo effects.

The enhanced pyrrolopyrimidine MIC when combined with betatide could be due to the enhanced uptake as betatide also affects the membrane via its cell penetrating peptide region [9]. In order to explore this further we combined compound **5** with commercial antibiotics targeting the membrane (cefotixin, methicillin, and only the cell penetrating part of betatide) or the bacterial translational machinery (gentamycin and clindamycin). We found a combinatory effect with some, but not all, of both types of antibiotics (data are not shown). This suggests that the combinatory effects between the fused pyrimidines and betatide are due to intracellular mechanisms and not an increased uptake of the compounds. Further studies to explore the molecular mechanisms behind the combinatory effects are ongoing.

## 3. Experimental Section

### 3.1. Chemicals and Materials

All the reagents and solvents used were purchased from Merck (Rahway, NJ, USA), VWR (Sugar Land, TX, USA) or Alpha Aesar (Ward Hill, MA, USA) and used without further purification. Compound **3** was previously prepared by Kaspersen et al. [45]. The reactions that were sensitive to moisture or oxygen were conducted under an N_2_ atmosphere using oven-dried glassware and solvents that were dried over molecular sieves for 24 h or collected from an MBraun SPS-800 solvent purifier. For flash chromatography, different stationary phases were used: for normal phase, silica-gel (40–63 µm particle size) purchased from VWR, prepackaged cartridges (Biotage Sfär Silica D Duo 60 µm and Biotage Sfär Silica HC D, 20 µm, Biotage, Uppsala, Sweeden, and Alumina, neutral, Brockmann I (58 Å pore size) that was purchased from Merck. For reversed phase flash chromatography prepackaged cartridges from Biotage (Biotage Sfär C18 D, 100 Å, 30 µm, 12 g) were used.

### 3.2. Analysis and Characterization

Accurate mass determination in positive and negative mode was performed on a “Synapt G2-S” Q-TOF instrument from Waters^TM^ (Milford, MA, USA). The samples were ionized using an ASAP probe (APCI). No chromatographic separation was done prior to the mass analysis. The calculated exact mass and spectra processing was done by Waters^TM^ Software (Masslynx V4.1 SCN871). NMR spectra were recorded using the Bruker DPX 400 MHz and 600 MHz Avance III HD NMR spectrometers. Chemical shifts (δ) are recorded in parts per million relative to TMS (δ_H_ = 0.00, δ_C_ = 0.00), CDCl_3_ (δ_H_ = 7.26, δ_C_ = 77.16), or DMSO-*d_6_* (δ_H_ = 2.50, δ_C_ = 39.52), and coupling constants (*J*) are measured in hertz (Hz). Purity analyses: HPLC purity analyses were performed on an Agilent 1260 series instrument with an ACE Excel 5 C18 column (4.6 mm × 150 mm, d_p_ = 5 µm) with a flow of 1 mL/min, UV monitoring at 320 nm, and with Agilent Chemstation as the software. Elution: 10 min linear gradient of MeCN/H_2_O (35:65) to MeCN/H_2_O (70:30) followed by 5 min linear gradient of MeCN/H_2_O (70:30) to MeCN/H_2_O (100:0). The enantiomeric excess of the intermediate **34** was controlled by HPLC using an Agilent 1100 series system detecting at 254 nM and using a Chiracel OD column (4.6 mm × 250 mm, d_p_ = 10 µm), mobile phase: *n*-hexane/2-propanol, 87:13, flow rate 1.0 mL/min; t_R_ = 8.15 min, t_S_ = 9.47 min, R_s_ = 1.7.

### 3.3. Synthetic Protocols

#### 3.3.1. Amination of Fused Pyrimidines (General Procedure A)

The following procedure is adapted from Kaspersen et al. [45]. The pyrrolo- or thienopyrimidine (100–200 mg) was flushed with N_2_ three times and dissolved in degassed *n*-BuOH (1–2 mL). DIPEA (0–6 equiv.) and amine (3 equiv.) were then added, and the reaction was stirred under an N_2_ atmosphere at an oil bath temperature of 140 °C for 18–24 h. The solution was then cooled to an ambient temperature and the solvent was removed in vacuo, facilitated by the addition of toluene. The resulting solid was suspended between EtOAc (25 mL) and H_2_O (25 mL) and separated. The aqueous layer was extracted with EtOAc (3 × 25 mL) and the combined organic layers were washed with brine (25 mL), dried over anhydrous Na_2_SO_4_, filtered, and concentrated in vacuo. The material was then purified by flash chromatography as described for each specific compound.

#### 3.3.2. Regioselective Suzuki Cross-Coupling (General Procedure B)

To a mixture of (*R*) or (*S*)-*N*-(1-(4-bromophenyl)ethyl)-6-iodo-7-((2-(trimethylsilyl)ethoxy)methyl)-7*H*-pyrrolo [2,3-*d*] pyrimidin-4-amine (**34**) (100–300 mg), arylboronic acid (1 equiv.), Pd (dppf) Cl_2_ (0.05 equiv.), and K_2_CO_3_ (3 equiv.) under N_2_, degassed 1,4-dioxane (2 mL) and water (1 mL) were added. The mixture was stirred at 80 °C until full conversion as noted by TLC or ^1^H NMR, before it was filtered through a celite pad which was rinsed with CH_2_Cl_2_ (4 × 10 mL) and the mixture was concentrated to dryness under reduced pressure. The solids were dissolved in CH_2_Cl_2_ (20 mL) and water (20 mL) and separated. The aqueous phase was extracted with CH_2_Cl_2_ (3 × 20 mL) before the combined organic phase was dried over Na_2_SO_4_, filtered, and concentrated in vacuo. The crude material was purified by flash chromatography as specified for each compound.

#### 3.3.3. SEM-Deprotection (General Procedure C)

The following two-step procedure is adapted from Reiersølmoen et al. [46]. The SEM-protected pyrrolopyrimidine (50–100 mg) was flushed with N_2_ three times and dissolved CH_2_Cl_2_ and TFA (1–2 mL). The reaction was then stirred under an N_2_ atmosphere at ambient temperature for 3–4 h. The TFA was then removed in vacuo, and co-evaporated twice with MeOH (3 × 10 mL) to remove traces of TFA. The reaction mixture was then dissolved in 1,4-dioxane (5–10 mL) and sat. NaHCO_3_ solution was added. This suspension was stirred overnight, before the solvent was removed in vacuo. The solid was then distributed between EtOAc (10–20 mL) and H_2_O (10–20 mL) and separated. The aqueous layer was extracted with EtOAc (4 × 10–20 mL) and the combined organic layers were washed with brine (10–20 mL), dried over anhydrous Na_2_SO_4_, filtered, and concentrated in vacuo. The solid was then purified as described in the specific experimental section.

### 3.4. Preparation of Compounds ***1–21***

#### 3.4.1. (*R*)-4-(4-((1-Phenylethyl)amino)-7H-pyrrolo[2,3-d]pyrimidin-6-yl)phenol (**1**)

Compound **1** was prepared as described in General Procedure C, starting from **25** (43 mg, 92.7 μmol). Immobilization on celite and purification by gradient flash chromatography (C18 silica, MeCN/H_2_O, 1:9 to 3:2) resulted in 23 mg (70.8 μmol, 76%) of the desired product as a white solid, mp. 226.6–228.4 °C (decomp.), ${\left[\alpha \right]}_{D}^{20}=-290$ (c 0.50, EtOH), HPLC purity: 99%, t_R_ = 5.61 min; TLC (H_2_O/MeCN 3:2) *R_f_* = 0.30; ^1^H NMR (600 MHz, DMSO-*d_6_*) δ: 11.82 (s, 1H), 9.61 (s, 1H), 8.02 (s, 1H), 7.68 (d, *J* = 8.2 Hz, 1H), 7.63–7.57 (m, 2H), 7.45–7.40 (m, 2H), 7.33–7.27 (m, 2H), 7.22–7.16 (m, 1H), 6.88 (d, *J* = 2.2 Hz, 1H), 6.85–6.80 (m, 2H), 5.49 (*p*, *J* = 7.1 Hz, 1H), 1.53 (d, *J* = 7.0 Hz, 3H); ^13^C NMR (151 MHz, DMSO-*d_6_*) δ: 157.0, 154.7, 151.1, 145.6, 134.0, 128.1 (2C), 126.4, 126.05 (2C), 126.03 (2C), 123.0, 115.7 (2C), 103.9, 93.9, 48.7, 22.9; IR (neat, cm^−1^): 3114 (O-H), 1596 (N-H), 1497 (Ar C-H), 699 (Ar C-H); HRMS (TOF ES+, *m/z*): calcd. for C_20_H_19_N_4_O [M + H]^+^: 331.1559, found: 331.1565. This material was first reported by Traxler et al. [47].

#### 3.4.2. (*R*)-4-(4-((1-(*p*-Tolyl)ethyl)amino)-7*H*-pyrrolo[2,3-*d*]pyrimidin-6-yl)phenol (**2**)

Compound **2** was prepared as described in General Procedure C, starting from **26** (102 mg, 0.215 mmol). The crude material was purified by preparative LC (Agilent Prep C-18 150 × 21.2 mm, 5 μm column, 20 mL/min flow, MeCN/H_2_O, 30:70 to 70:30 (0–10 min linear gradient), followed by 70:30–100:0 (10–15 min linear gradient), 100 μL injection, λ = 320 nm, t_R_ = 8.139 min), resulting in 14 mg (40.4 μmol, 19%) of the desired product as a white solid, ${\left[\alpha \right]}_{D}^{20}=-266$ (c 0.50, EtOH), HPLC purity: 96%, t_R_ = 6.49 min. ^1^H NMR (600 MHz, DMSO-*d_6_*) δ: 11.80 (s, 1H), 9.60 (s, 1H), 8.01 (s, 1H), 7.62 (d, *J* = 8.3 Hz, 1H), 7.61–7.56 (m, 2H), 7.33–7.26 (m, 2H), 7.10 (m, 2H), 6.87 (d, *J* = 2.2 Hz, 1H), 6.85–6.79 (m, 2H), 5.44 (*p*, *J* = 7.2 Hz, 1H), 2.25 (s, 3H), 1.50 (d, *J* = 7.0 Hz, 3H); ^13^C NMR (151 MHz, DMSO-*d_6_*) δ: 156.9, 154.3, 151.1, 142.6, 135.3, 134.0, 128.7 (2C), 126.02 (2C), 125.97 (2C), 123.0, 115.7 (2C), 103.7, 94.6, 48.4, 22.9, 20.6; IR (neat, cm^−1^): 3136 (O-H), 1598 (N-H), 1498 (Ar C-H), 835 (Ar C-H); HRMS (TOF ES+, *m/z*): calcd. for C_21_H_21_N_4_O [M + H]^+^: 345.1715, found: 345.1719.

#### 3.4.3. (*R*)-4-(4-((1-(4-Chlorophenyl)ethyl)amino)-7*H*-pyrrolo[2,3-*d*]pyrimidin-6-yl)phenol (**4**)

Compound **4** was prepared as described in General Procedure C, starting from **27** (51 mg, 0.103 mmol). Extraction with EtOAc and H_2_O resulted in 37 mg (0.101 mmol, 98%) of the desired product as a light-yellow solid, mp. 216.6 °C (decomp.), ${\left[\alpha \right]}_{D}^{20}=-352$ (c 0.50, EtOH), HPLC purity: 96%, t_R_ = 7.81 min. ^1^H NMR (600 MHz, DMSO-*d_6_*) δ: 11.84 (s, 1H), 9.62 (s, 1H), 8.01 (s, 1H), 7.72 (d, *J* = 8.0 Hz, 1H), 7.63–7.58 (m, 2H), 7.46–7.41 (m, 2H), 7.38 –7.33 (m, 2H), 6.87 (d, *J* = 2.2 Hz, 1H), 6.86–6.80 (m, 2H), 5.45 (*p*, *J* = 7.2 Hz, 1H), 1.51 (d, *J* = 7.0 Hz, 3H); ^13^C NMR (151 MHz, DMSO-*d_6_*) δ: 157.0, 151.1, 144.8, 134.2, 130.9, 128.1 (2C), 128.0 (2C), 126.1 (2C), 122.9, 115.7 (2C), 103.9, 93.9, 48.2, 22.8; IR (neat, cm^−1^): 3129 (O-H), 1599 (N-H), 1495 (Ar C-H), 833 (Ar C-H); HRMS (TOF ES+, *m/z*): calcd. for C_20_H_18_N_4_OCl [M+H]^+^ 365.1169, found: 365.1169.

#### 3.4.4. (*R*)-4-(4-((1-(4-Bromophenyl)ethyl)amino)-7*H*-pyrrolo[2,3-*d*]pyrimidin-6-yl)phenol (**5**) from **28**

Compound **5** was prepared as described in General Procedure C, starting from **28** (42 mg, 0.078 mmol). The crude product was dissolved in MeOH (3 mL) and purified by gradient flash chromatography (C18 silica, MeOH/H_2_O, 1:1 to 9:1). TLC (MeOH/H_2_O, 9:1) *R_f_* = 0.48. This afforded 22 mg (0.054 mmol, 69%) as a white solid; HPLC purity: 96%, t_R_ = 7.37 min; ${\left[\alpha \right]}_{D}^{20}$= −352.0 (1.00, EtOH (96%)); ^1^H NMR (600 MHz, DMSO-*d_6_*) δ: 11.81 (s, 1H), 8.00 (s, 1H), 7.70 (d, *J* = 8.0 Hz, 1H), 7.59–7.57 (m, 2H), 7.51–7.48 (m, 2H), 7.39–7.38 (m, 2H), 6.84 (s, 1H), 6.81–6.79 (m, 2H), 5.43 (*p*, *J* = 7.2 Hz, 1H), 1.51 (d, *J* = 7.0 Hz, 3H), OH-signal not seen; ^13^C NMR (150 MHz, DMSO-*d_6_*) δ: 158.2 154.5, 151.2, 150.9, 145.3, 134.5, 131.0 (2C), 128.3 (2C), 126.0 (2C), 122.0, 119.3, 116.0 (2C), 103.9, 93.5, 48.3, 22.8; IR (neat, cm^−1^): 3328 (N-H), 3122 (O-H), 1596 (N-H), 834 (Ar C-H); HRMS (ASAP-TOF, *m*/*z*): calcd. for C_20_H_18_N_4_O^79^Br, 409.0664 [M + H]^+^, found 409.0670.

#### 3.4.5. (*R*)-4-(4-((1-(4-Bromophenyl)ethyl)amino)-7*H*-pyrrolo[2,3-*d*]pyrimidin-6-yl)phenol (**5**) from **35**


The following procedure is adapted from Kaspersen et al. [45]. Compound **35** (38 mg, 69.4 μmol) was dissolved in CH_2_Cl_2_ (1 mL) and stirred at 0 °C under an N_2_ atmosphere. BBr_3_ (1 M, 700 μL) was added dropwise over 1 hour, and the reaction was stirred for a further 1.5 h at 22 °C. The reaction was then cooled to 0 °C and quenched by the addition of H_2_O (4 mL) and saturated NaHCO_3_ solution (1 mL). The pH was adjusted to 4 with HCl (2 M) and saturated NaHCO_3_ solution. The solution was then distributed between EtOAc (25 mL) and H_2_O (25 mL) and separated. The aqueous phase was extracted with EtOAc (3 × 25 mL) and the combined organic layers were washed with brine (25 mL), dried over anhydrous Na_2_SO_4_, filtered, and concentrated in vacuo. The resulting crude product was dissolved in 1,4-dioxane (10 mL) and saturated NaHCO_3_ solution (1 mL) was added. The solution was stirred under an N_2_ atmosphere overnight, before it was concentrated in vacuo and suspended in H_2_O (25 mL) and EtOAc (25 mL). The pH in the aqueous phase was adjusted to 5 with HCl (2 M) and saturated NaHCO_3_ solution. The phases were separated, and the aqueous layer was extracted with EtOAc (3 × 25 mL). The combined organic phases were washed with brine, dried over Na_2_SO_4_, filtered, and concentrated in vacuo. The resulting crude was immobilized on Celite and purified by flash chromatography (C18-silica, MeCN/H_2_O cont. 0.1% NEt_3_ in H_2_O, 3:2, *R_f_* = 0.29), resulting in 12 mg (2.86 μmol, 41%) of the desired product as a light-yellow solid. The ^1^H NMR spectroscopic data confirmed with that which is reported above [45].

#### 3.4.6. *(R)-*4-(4-((1-(4-Iodophenyl)ethyl)amino)-7*H*-pyrrolo[2,3-*d*]pyrimidin-6-yl)phenol (**6**)

Compound **6** was prepared as described in General Procedure C, starting from **29** (51 mg, 86.1 μmol). Immobilization on celite and purification by gradient flash chromatography (C18 silica, MeCN/H_2_O, 1:9 to 3:2) resulted in 22 mg (48.4 μmol, 56%) of the desired product as a light-yellow solid, mp. 254.9-256.0 °C (decomp.), TLC (MeCN/H_2_O 3:2) *R_f_* = 0.24; ${\left[\alpha \right]}_{D}^{20}=-356$ (c 0.50, EtOH), HPLC purity: 97%, t_R_ = 7.85 min. ^1^H NMR (600 MHz, DMSO-*d_6_*) δ: 11.84 (s, 1H), 9.62 (s, 1H), 8.00 (s, 1H), 7.70 (d, *J* = 8.0 Hz, 1H), 7.68–7.63 (m, 2H), 7.63–7.57 (m, 2H), 7.26–7.21 (m, 2H), 6.86 (d, *J* = 2.1 Hz, 1H), 6.85–6.76 (m, 2H), 5.40 (*p*, *J* = 7.2 Hz, 1H), 1.50 (d, *J* = 6.9 Hz, 3H); ^13^C NMR (151 MHz, DMSO-*d_6_*) δ: 157.0, 151.3, 151.1, 145.7, 136.9 (2C), 134.2, 128.5 (2C), 126.1 (2C), 122.9, 115.7 (2C), 103.9, 93.9, 92.0, 48.4, 22.7; IR (neat, cm^−1^): 3139 (O-H), 1598 (N-H), 1498 (Ar C-H), 834 (Ar C-H); HRMS (TOF ES+, *m/z*): calcd. for C_20_H_18_N_4_OI [M + H]^+^: 457.0525, found: 457.0534.

#### 3.4.7. *(R)*-4-(4-((1-(3-Bromophenyl)ethyl)amino)-7*H*-pyrrolo[2,3-*d*]pyrimidin-6-yl)phenol (**7**)

Compound **7** was prepared as described in General Procedure C, starting from **30** (103 mg, 0.190 mmol). Immobilization on celite and purification by gradient flash chromatography (silica, EtOAc/*n*-pentane 0:1 to 1:1) resulted in 49 mg (0.199 mmol, 63%) of the desired product as an off-white solid, decomp. at 237.5–240.1 °C, TLC (EtOAc/*n-*pentane 1:1) *R_f_* = 0.39; ${\left[\alpha \right]}_{D}^{20}=-228$ (c 0.5, EtOH), HPLC purity: 98%, t_R_ = 7.24 min. ^1^H NMR (600 MHz, DMSO-*d_6_*) δ: 11.86 (s, 1H), 9.62 (s, 1H), 8.02 (s, 1H), 7.74 (d, *J* = 8.1 Hz, 1H), 7.64–7.58 (m, 3H), 7.45–7.40 (m, 1H), 7.40–7.37 (m, 1H), 7.27 (t, *J* = 7.8 Hz, 1H), 6.86 (d, *J* = 2.2 Hz, 1H), 6.86–6.80 (m, 2H), 5.48–5.43 (m, 1H), 1.51 (d, *J* = 7.0 Hz, 3H); ^13^C NMR (151 MHz, DMSO-*d_6_*) δ: 157.0, 154.5, 151.1, 148.8, 134.3, 130.5, 129.3, 128.7, 126.1, 125.3, 122.9, 121.6, 115.7, 103.9, 93.8, 48.4, 22.9; IR (neat, cm^−1^): 3111 (O-H), 1596 (N-H), 1498 (Ar C-H), 521 (C-Br); HRMS (TOF ES+, *m/z*): calcd. for C_20_H_18_N_4_O^79^Br [M + H]^+^: 409.0664, found: 409.0667.

#### 3.4.8. 4-(4-((4-Bromobenzyl)amino)-7*H*-pyrrolo[2,3-*d*]pyrimidin-6-yl)phenol (**8**)

Compound **8** was prepared as described in General Procedure C starting from **31** (111 mg, 0.211 mmol). Immobilization on celite and purification by gradient flash chromatography (C18 silica, MeCN/H_2_O, 1:9 to 1:0) gave the desired product in a yield of 11 mg (28.5 µmol, 13%) as an off-white solid, TLC (MeCN/H_2_O 3:2) *R_f_* = 0.33; HPLC purity: 96%, t_R_ = 6.65 min. ^1^H NMR (600 MHz, DMSO-*d_6_*) δ: 11.87 (s, 1H), 9.61 (s, 1H), 8.07 (s, 1H), 7.94 (t, *J* = 6.1 Hz, 1H), 7.60–7.58 (m, 2H), 7.52–7.48 (m, 2H), 7.32–7.30 (m, 2H), 6.84–6.81 (m, 2H), 6.76 (d, *J* = 2.2 Hz, 1H), 4.68 (d, *J* = 6.1 Hz, 2H); ^13^C NMR (151 MHz, DMSO-*d_6_*) δ: 157.0, 155.3, 151.2, 151.2, 140.0, 134.3, 131.1 (2C), 129.4 (2C), 126.1 (2C), 122.9, 119.5, 115.7 (2C), 103.9, 93.6, 42.6; IR (neat, cm^−1^): 3421 (N-H), 3109 (O-H), 1599 (N-H), 1500 (Ar C-H), 800 (Ar C-H); HRMS (TOF ES+, *m/z*): calcd. for C_19_H_16_N_4_O^79^Br [M + H]^+^ 395.0507, found: 395.0504.

#### 3.4.9. (*S*)-4-(4-((1-(4-Bromophenyl)ethyl)amino)-7*H*-pyrrolo[2,3-*d*]pyrimidin-6-yl)phenol (**9**)

Compound **9** was prepared as described in General Procedure C starting from **32** (98.8 mg, 0.183 mmol). Immobilization on celite and purification by gradient flash chromatography (C18 silica, MeCN/H_2_O, 1:9 to 1:0). This gave 49 mg (48.7 mg, 0.199 mmol, 65%) of a white solid, mp. 253.1–254.5 °C (decomp.); TLC (MeOH/H_2_O, 9:1) *R_f_* = 0.48; ^1^H NMR (600 MHz, DMSO-*d_6_*) δ: 11.85 (s, 1H), 9.63 (s, 1H), 8.02 (s, 1H), 7.72 (d, *J* = 8.0 Hz, 1H), 7.61 (d, *J* = 8.1 Hz, 2H), 7.49 (d, *J* = 8.5 Hz, 2H), 7.38 (d, *J* = 8.5 Hz, 2H), 6.87 (s, 1H), 6.84 (d, *J* = 8.0 Hz, 2H), 5.43 (p, *J* = 7.2 Hz, 1H), 1.51 (d, *J* = 7.0 Hz, 3H); ^13^C NMR (151 MHz, DMSO*-d_6_*) δ: 157.0, 154.6, 151.3, 151.1, 145.2, 134.2, 131.0 (2C), 128.4 (2C), 126.1 (2C), 122.9, 119.3, 115.7 (2C), 103.9, 93.9, 48.6, 22.8^.^; IR (neat, cm^−1^): 2975 (C-H), 1547 (N-H), 834 (Ar C-H), 548 (C-Br); HRMS (TOF ES+, *m/z*): calcd. for C_20_H_18_ N_4_O^79^Br [M + H]^+^: 409.0664, found: 409.0664. The ^1^H NMR shifts are found at higher field than that reported for the corresponding HBr-salt [48].

#### 3.4.10. *(R)*-4-(4-((1-(4-Bromophenyl)ethyl)(methyl)amino)-7*H*-pyrrolo [2,3-*d*]pyrimidin-6-yl)phenol (**10**)

Compound **10** was prepared as described in General Procedure C starting from **33** (52 mg, 93.9 µmol). Immobilization on celite and purification by gradient flash chromatography (silica, MeOH/CH_2_Cl_2_, 0:100 to 5:95) yielded a yellow solid, which after a wash with toluene (5 mL) resulted in the desired product with a yield of 59% (18.8 mg, 44.4 µmol) as an off-white solid, TLC (MeOH/CH_2_Cl_2_ 3:2) *R_f_* = 0.24; ${\left[\alpha \right]}_{D}^{20}=-10.0$ (c 0.50, DMF); HPLC purity: 98.5%, t_R_ = 9.57 min; ^1^H NMR (600 MHz, DMSO*-d_6_*) δ: 12.00 (s, 1H), 9.58 (s, 1H), 8.13 (s, 1H), 7.72–7.65 (m, 2H), 7.58–7.49 (m, 2H), 7.32–7.24 (m, 2H), 6.88 (d, *J* = 2.2 Hz, 1H), 6.84–6.76 (m, 2H), 6.39 (s, 1H), 3.06 (s, 3H), 1.58 (d, *J* = 7.0 Hz, 3H); ^13^C NMR (151 MHz, DMSO-*d_6_*) δ: 157.0, 156.2, 152.8, 150.5, 141.3, 133.9, 131.3 (2C), 129.2 (2C), 126.3 (2C), 122.6, 120.0, 115.6 (2C), 103.6, 96.8, 51.9, 40.1, 31.5, 16.2; IR (neat, cm^−1^): 3115 (O-H), 1568 (N-H), 1490 (Ar C-H), 827 (Ar C-H); HRMS (TOF ES+, *m/z*): calcd. For C_21_H_20_N_4_O^79^Br [M + H]^+^: 423.0820, found: 423.0826.

#### 3.4.11. (*S*)-*N*-(1-(4-Bromophenyl)ethyl)-6-(4-methoxyphenyl)-7*H*-pyrrolo[2,3-*d*]pyrimidin-4-amine (**11**)

Compound **11** was prepared as described in General Procedure C starting from **35** (120 mg, 0.217 mmol). The crude product was purified twice by C18 silica flash chromatography (first MeOH/EtOAc, 3:1, then: acetone/H_2_O, 1:1, TLC (C18 silica acetone/H_2_O, 1:1) *R_f_* = 0.68). This gave 33 mg (0.078 mmol, 36%) as an off-white solid; HPLC purity: 98%, t_R_ = 9.86 min; ${\left[\alpha \right]}_{D}^{20}$ = +288.2 (1.00, EtOH (100%); ^1^H NMR (600 MHz, DMSO-*d_6_*) δ: 11.932-11.929 (m, 1H), 8.02 (s, 1H), 7.76 (d, 1H, *J* = 8.1 Hz), 7.73-7.71 (m, 2H), 7.50–7.48 (m, 2H), 7.39–7.37 (m, 2H), 7.03-7.01 (m, 2H), 6.94–6.93 (br d, 1H, *J* = 1.6 Hz), 5.43 (quint., 1H, *J* = 7.2 Hz), 3.80 (br s, 3H), 1.51 (d, 3H, *J* = 7.0 Hz); ^1^H NMR is in accordance with that reported at 400 MHz [48].

#### 3.4.12. (*R*)-*N*-(1-(4-Bromophenyl)ethyl)-6-phenyl-7*H*-pyrrolo[2,3-*d*] pyrimidin-4-amine (**12**)

Compound **12** was prepared as described in General Procedure C starting from **36** (63 mg, 0.120 mmol). The crude product was purified twice by C18 silica gradient flash chromatography (first: acetone/MeOH, 0:100 to 1:4 acetone/MeOH, then: MeOH/H_2_O, 1:1 to 9:1. TLC (C-18 silica, acetone/MeOH, 1:4) *R_f_* = 0.50). This gave 32 mg (0.082 mmol, 68%) as a white solid; HPLC purity >99% t_R_ = 10.03 min; ${\left[\alpha \right]}_{D}^{20}=$−369.0 (1.00, EtOH abs); ^1^H NMR (600 MHz, DMSO-*d_6_*) δ: 12.05 (s, 1H), 8.05 (s, 1H), 7.84 (d, *J* = 7.9 Hz, 1H), 7.80–7.78 (m, 2H), 7.51–7.48 (m, 2H), 7.46-7.43 (m, 2H), 7.39-7.37 (m, 2H), 7.31–7.28 (m, 1H), 7.09 (d, *J* = 1.6 Hz, 1H), 5.44 (quint., *J* = 7.4 Hz, 1H), 1.52 (d, *J* = 7.1 Hz, 3H); ^13^C NMR (150 MHz, DMSO-*d_6_*) δ: 154.9, 151.7, 151.6, 145.1, 133.5, 131.7, 131.0 (2C), 129.0 (2C), 128.3 (2C), 127.3, 124.5 (2C), 119.4, 103.9, 96.0, 48.3, 22.7; IR (neat, cm^−1^): 3415 (N-H), 3114 (C-H), 1595 (N-H), 1475 (Ar C-H), 751 (C-H); HRMS (ASAP-TOF, *m*/*z*): calcd. for C_20_H_18_N_4_^79^Br, 393.0715 [M + H]^+^), found 393.0715.

#### 3.4.13. (*R*)-*N*-(1-(4-Bromophenyl)ethyl)-6-(4-fluorophenyl)-7*H*-pyrrolo[2,3-*d*] pyrimidin-4-amine (**13**)

Compound **13** was prepared as described in General Procedure C starting from **37** (63 mg, 0.116 mmol). The crude product was purified by gradient flash chromatography (C18 silica, MeOH/H_2_O, 1:1 to 9:1). TLC (MeOH/H_2_O, 9:1, *R_f_* = 0.20). This gave 37 mg (0.090 mmol, 78%) as a white solid; HPLC purity: 99%, t_R_ = 10.310 min; ${\left[\alpha \right]}_{D}^{20}$= −317.0 (1.00, EtOH abs); ^1^H NMR (600 MHz, CDCl_3_) δ: 12.82 (s, 1H), 8.31 (s, 1H), 7.75–7.69 (m, 2H), 7.50–7.45 (m, 2H), 7.35–7.30 (m, 2H), 7.21–7.14 (m, 2H), 6.53 (d, *J* = 1.9 Hz, 1H), 5.52 (*p*, *J* = 7.0 Hz, 1H), 5.28 (s, 1H), 1.66 (d, *J* = 6.9 Hz, 3H); ^13^C NMR (150 MHz, CDCl_3_) δ:162.7 (d, *J* = 247.6 Hz), 155.5, 152.0, 151.6, 143.3, 135.3, 131.9 (2C), 128.5 (d, *J* = 3.2 Hz), 128.0 (2C), 127.3 (d, *J* = 8.8 Hz, 2C), 121.2, 116.3 (d, *J* = 22.0 Hz, 2C), 104.6, 94.5, 49.9, 23.0; ^19^F NMR (376 MHz, CDCl_3_, C_6_F_6_) δ: −116.3 (s); IR (neat, cm^−1^): 3428 (N-H), 2976 (C-H), 1596 (N-H), 1496 (Ar C-H), 834 (Ar C-H); HRMS (ASAP-TOF, *m*/*z*): calcd. for C_20_H_17_N_4_F^79^Br, 411.0621 [M + H]^+^), found 411.0621.

#### 3.4.14. (*R*)-*N*-(1-(4-Bromophenyl)ethyl)-6-(4-nitrophenyl)-7*H*-pyrrolo [2,3-*d*] pyrimidin-4-amine (**14**)

Compound **14** was prepared as described in General Procedure C starting from **38** (185 mg, 0.325 mmol). The crude product was dissolved in DMF (2 mL) at 75 °C and water was added dropwise until saturation, which produced precipitation upon cooling. This afforded 73 mg (0.167 mmol, 51%) of a yellow-orange solid; HPLC purity: 96%, t_R_ = 10.37 min; ${\left[\alpha \right]}_{D}^{20}$= −635.0 (1.00, DMSO); ^1^H NMR (600 MHz, DMSO-*d_6_*) δ: 12.37 (s, 1H), 8.32–8.30 (m, 2H), 8.11 (s, 1H), 8.07 (d, *J* = 7.9 Hz, 1H), 8.03–8.00 (m, 2H), 7.51–7.49 (m, 2H), 7.39–7.37 (m, 3H), 5.45 (*p*, *J* = 7.1 Hz, 1H), 1.53 (d, *J* = 7.0 Hz, 3H); ^13^C NMR (150 MHz, DMSO-*d_6_*) δ: 155.3, 153.0, 152.3, 145.7, 144.8, 138.0, 131.2, 131.1 (2C), 128.3 (2C), 125.0 (2C), 124.5 (2C), 119.5, 104.2, 100.1, 48.4, 22.6; IR (neat, cm^−1^): 3407 (N-H), 2973 (C-H), 1601 (N-H), 1536 (N-O), 1489 (Ar C-H), 828 (Ar C-H); HRMS (ASAP-TOF, *m*/*z*): calcd. for C_20_H_17_N_5_O_2_^79^Br, 438.0566 [M + H]^+^, found 438.0563.

#### 3.4.15. (*R*)-6-(4-Aminophenyl)-*N*-(1-(4-bromophenyl)ethyl)-7*H*-pyrrolo[2,3-*d*]pyrimidin-4-amine (**15**)

To a mixture the nitro derivative **14** (40 mg, 0.091 mmol), NH_4_Cl (44 mg, 0.823 mmol) and Fe-powder (15 mg, 0.276 mmol) under a N_2_ atmosphere, water (0.9 mL), and EtOH (96%, 2.1 mL) were added. The mixture was stirred at 78 °C for 5 hours before full conversion was observed by TLC. The mixture was filtered through a celite pad, which was washed with EtOAc (75 mL) and MeOH (100 mL), and the solution was dried in vacuo. The solids were added to water (20 mL) and EtOAc (50 mL), and the aqueous layer was adjusted to pH 9 with NaHCO_3_ before the layers were separated, and the aqueous layer was extracted with additional EtOAc (3 × 20 mL). The combined organic phase was dried over Na_2_SO_4_, filtered, and concentrated in vacuo. The crude product was purified by gradient flash chromatography (C18 silica, MeOH/H_2_O, 1:1 to 9:1). TLC (MeOH/H_2_O, 9:1) *R_f_* = 0.39. This afforded 17 mg (0.042 mmol, 45%) as an off-white solid; HPLC purity: 96%, t_R_ = 7.34 min; ${\left[\alpha \right]}_{D}^{20}$= −315 (1.00, acetone); ^1^H NMR (600 MHz, DMSO-*d_6_*) δ: 11.71 (br s, 1H), 7.98 (s, 1H), 7.66 (d, *J* = 8.1 Hz, 1H), 7.50–7.45 (m, 4H), 7.38–7.37 (m, 2H), 6.76 (d, *J* = 2.1 Hz, 1H), 6.62–6.60 (m, 2H), 5.42 (*p*, *J* = 7.2 Hz, 1H), 1.50 (d, *J* = 7.0 Hz, 3H) (NH_2_-signal not observed); ^13^C NMR (150 MHz, DMSO-*d_6_*) δ: 154.2, 151.1, 150.5, 148.4, 145.3, 135.1, 131.0 (2C), 128.3 (2C), 125.7 (2C), 119.4, 119.3, 114.0 (2C), 104.0, 92.5, 48.3, 22.8; IR (neat, cm^−1^): 3342 (N-H), 3214 (N-H), 2923 (C-H), 1594 (N-H), 1476 (Ar C-H), 825 (Ar C-H); HRMS (ASAP-TOF, *m*/*z*): calcd. for C_20_H_19_N_5_^79^Br, 408.0824 [M + H]^+^, found 408.0829.

#### 3.4.16. Methyl (*R*)-4-(4-((1-(4-bromophenyl)ethyl)amino)-7*H*-pyrrolo[2,3-*d*] pyrimidin-6-yl)be-nzoate (**16**)

Compound **16** was prepared as described in General Procedure C starting from **39** (74 mg, 0.127 mmol). The crude product was suspended in acetone (3 mL) and centrifuged at 4400 rpm for 10 min before the supernatant was removed. This gave 31 mg (0.069 mmol, 54%) as a white powder; HPLC purity: 99%, t_R_ = 10.03 min; ${\left[\alpha \right]}_{D}^{20}$= −467.0 (c 1.00, DMSO); ^1^H NMR (600 MHz, DMSO-*d_6_*) δ: 12.23 (d, *J* = 2.2 Hz, 1H), 8.08 (s, 1H), 8.03–8.01 (m, 2H), 7.97 (d, *J* = 8.0 Hz, 1H), 7.93–7.91 (m, 2H), 7.51–7.49 (m, 2H), 7.39–7.37 (m, 2H), 7.28 (d, *J* = 2.1 Hz, 1H), 5.45 (*p*, *J* = 7.2 Hz, 1H), 3.87 (s, 3H), 1.52 (d, *J* = 7.0 Hz, 3H); ^13^C NMR (150 MHz, DMSO-*d_6_*) δ: 165.9, 155.2, 152.5, 152.0, 144.9, 136.2, 132.2, 131.1 (2C), 129.9 (2C), 128.3 (2C), 127.7, 124.4 (2C), 119.4, 104.0, 98.4, 52.1, 48.4, 22.7; IR (neat, cm^−1^): 3369 (N-H), 2969, 2949 (C-H), 1696 (C=O), 1597 (N-H), 1475 (Ar C-H), 823 (Ar C-H), 602 (C-Br); HRMS (ASAP-TOF, *m*/*z*): calcd. for C_22_H_20_N_4_O_2_^79^Br, 451.0770 [M + H]^+^, found 451.0770.

#### 3.4.17. (*R*)-4-(4-((1-(4-Bromophenyl)ethyl)amino)-7*H*-pyrrolo[2,3-*d*]pyrimidin-6-yl)benzene-sulfonamide (**17**)

Compound **17** was prepared as described in General Procedure C starting from **40** (122 mg, 0.202 mmol). The crude material was purified by gradient flash chromatography (C18 silica, acetone/H_2_O, 0:100 to 1:1). TLC (acetone/H_2_O, 1:1) *R_f_* = 0.25. This afforded 67 mg (0.142 mmol, 70%) as a white powder; HPLC purity: 96%, t_R_ = 6.875 min; ${\left[\alpha \right]}_{D}^{20}$ = −367 (1.00, acetone); ^1^H NMR (600 MHz, DMSO-*d_6_*) δ: 12.22 (d, 1H, *J* = 2.2 Hz), 8.09 (s, 1H), 7.98 (d, 1H, *J* = 8.0 Hz), 7.96–7.94 (m, 2H), 7.89–7.86 (m, 2H), 7.52-7.49 (m, 2H), 7.40–7.37 (m, 4H), 7.25 (d, 1H, *J* = 2.0 Hz), 5.45 (quint., 1H, *J* = 7.2 Hz), 1.53 (d, 3H, *J* = 7.0 Hz); ^13^C NMR (150 MHz, DMSO-*d_6_*) δ: 155.2, 152.4, 151.9, 145.0, 142.2, 134.8, 132.0, 131.1 (2C), 128.3 (2C), 126.4 (2C), 124.6 (2C), 119.4, 104.0, 98.2, 48.4, 22.7; IR (neat, cm^−1^): 3376 (N-H), 2923 (C-H), 1593 (N-H), 1477 (Ar C-H), 1331 (S=O), 827 (Ar C-H), 621 (C-Br); HRMS (ASAP-TOF, *m*/*z*): calcd. for C_20_H_19_N_5_O_2_S^79^Br, 472.0443 [M + H]^+^), found 472.0444.

#### 3.4.18. (*R*)-2-(4-((1-(4-Bromophenyl)ethyl)amino)-7*H*-pyrrolo[2,3-*d*]pyrimidin-6-yl)phenol (**18**)

Compound **18** was prepared as described in General Procedure C starting from **41** (48 mg, 0.089 mmol). The crude product was purified by gradient flash chromatography (C18 silica, acetone/H_2_O, 1:3 to 4:1). TLC (C18 silica, acetone/H_2_O,4:1) *R_f_* = 0.77. This afforded 24 mg (0.059 mmol, 66%) of a white powder; HPLC purity: 97%, t_R_ =8.43 min; ${\left[\alpha \right]}_{D}^{20}$ = −367.0 (1.00, EtOH); ^1^H NMR (600 MHz, DMSO-*d_6_*) δ: 11.58 (br s, 1H), 10.06 (br s, 1H), 8.02 (s, 1H), 7.87 (d, *J* = 8.0 Hz, 1H), 7.71 (dd, *J* = 7.8, 1.7 Hz, 1H), 7.50–7.47 (m, 2H), 7.39-7.37 (m, 2H), 7.28 (s, 1H), 7.11 (ddd, *J* = 8.1, 7.2, 1.6 Hz, 1H), 6.98 (dd, *J* = 8.2, 1.2 Hz, 1H), 6.87 (td, *J* = 7.5, 1.2 Hz, 1H), 5.45 (quint., *J* = 7.2 Hz, 1H), 1.51 (d, *J* = 7.1 Hz, 3H); ^13^C NMR (150 MHz, DMSO-*d_6_*) δ: 154.8, 154,4, 151.3, 150.6, 145.3, 131.0 (2C), 130.7, 128.4 (2C), 127.9, 126.9, 119.3 (2C)*, 118.5, 116.3, 103.7, 99.4, 48.3, 22.6, *2 overlapping signals revealed by HMBC; IR (neat, cm^−1^): 3404 (N-H), 3046 (O-H), 2969 (C-H), 1596 (N-H), 1470 (Ar C-H), 822 (Ar C-H), 582 (C-Br); HRMS (ASAP-TOF, *m*/*z*): calcd. for C_20_H_18_N_4_O^79^Br, 409.0664 [M + H]^+^, found 409.0665.

#### 3.4.19. (*R*)-3-(4-((1-(4-Bromophenyl)ethyl)amino)-7*H*-pyrrolo[2,3-*d*]pyrimidin-6-yl)phenol (**19**)

Compound **19** was prepared as described in General Procedure C starting from **42** (47 mg, 0.087 mmol). The crude product was purified by gradient flash chromatography (C18 silica, acetone/H_2_O, 1:1 to 2:1). TLC (silica-gel, acetone/*n*-pentane 1:2) *R_f_* = 0.24. This gave 28 mg (0.068, 78%) as a white solid; HPLC purity: 97%, t_R_ =7.65 min; ${\left[\alpha \right]}_{D}^{20}$ = −296.0 (c 1.00, EtOH); ^1^H NMR (600MHz, DMSO-*d_6_*) δ: 11.96 (br d, *J* = 2.3 Hz, 1H), 9.54 (s, 1H), 8.04 (s, 1H), 7.82 (d, *J* = 8.0 Hz, 1H), 7.50–7.48 (m, 2H), 7.39–7.37 (m, 2H), 7.23–7.21 (m, 2H), 7.164–7.157 (m, 1H), 7.00 (br d, *J* = 2.2 Hz 1H), 6.72-6.70 (m, 1H), 5.44 (quint., *J* = 7.2 Hz, 1H), 1.52 (d, *J* = 7.0 Hz, 3H); ^13^C NMR (150 MHz, DMSO-*d_6_*) δ: 157.7, 154.9, 151.6, 151.4, 145.1, 133.7, 133.0, 131.0 (2C), 129.9, 128.3 (2C), 119.3, 115.5, 114.5, 111.5, 95.9, 48.3, 30.7, 22.7; IR (neat, cm^−1^): 3119 (O-H), 2965 (C-H), 1597 (N-H), 1475 (Ar C-H), 822 (Ar C-H), 687 (C-Br); HRMS (ASAP-TOF, *m*/*z*): calcd. for C_20_H_18_N_4_O^79^Br, 409.0664 [M + H]^+^, found 409.0670.

#### 3.4.20. *(R)-N*-(1-(4-Bromophenyl)ethyl)-6-(4-methoxyphenyl)thieno [2,3-*d*] pyrimidin-4-amine (**20**)

4-Chloro-6-(4-methoxyphenyl)thieno [2,3-*d*] pyrimidine (**44**) (228 mg, 0.823 mmol) and *(R)-*1-(4-bromophenyl)ethan-1-amine (365 µL, 2.50 mmol) were reacted as described in General Procedure A. The resulting crude material was purified by gradient flash chromatography (silica, EtOAc/*n-*pentane, 0:1 to 1:1), resulting in 187 mg (0.424 mmol, 52%) of the desired product as a yellow solid, mp. 107.1–109.4 °C, TLC (EtOAc/*n*-pentane 4:1) *R_f_* = 0.29; ${\left[\alpha \right]}_{D}^{20}=-416$ (c 0.50, EtOH), HPLC purity >99%, t_R_ = 13.93 min. ^1^H NMR (600 MHz, DMSO-*d_6_*) δ: 8.25 (s, 1H), 8.18 (d, *J* = 7.7 Hz, 1H), 8.02 (s, 1H), 7.66–7.60 (m, 2H), 7.54–7.48 (m, 2H), 7.41–7.35 (m, 2H), 7.11–7.05 (m, 2H), 5.45 (t, *J* = 7.2 Hz, 1H), 3.82 (s, 3H), 1.55 (d, *J* = 7.0 Hz, 3H). ^13^C NMR (151 MHz, DMSO-d_6_) δ: 164.7, 159.6, 155.5, 153.5, 144.2, 138.4, 131.2 (2C), 128.3 (2C), 127.0 (2C), 125.8, 119.6, 117.6, 114.8 (2C), 113.8, 55.3, 48.7, 22.4; IR (neat, cm^−1^): 3297 (N-H), 1607 (N-H), 1523 (Ar C-H), 825 (Ar C-H); HRMS (TOF ES+, *m/z*): calcd. for C_21_H_19_BrN_3_OS [M + H]^+^: 440.0432, found: 440.0436.

#### 3.4.21. (*R*)-4-(4-((1-(4-Bromophenyl)ethyl)amino)thieno[2,3-*d*]pyrimidin-6-yl)phenol (**21**)

(R)-*N*-(1-(4-Bromophenyl)ethyl)-6-(4-methoxyphenyl)thieno[2,3-*d*]pyrimidin-4-amine (100 mg, 0.228 mmol) was flushed three times with N_2_ and dissolved in dry CH_2_Cl_2_ (2 mL). The reaction was cooled to 0 °C and BBr_3_ in DCM (1 M, 2.5 mL) was added dropwise over 1 h. After a further 3.5 h, the reaction was quenched with H_2_O (7.5 mL) and sat. NaHCO_3_ solution (5 mL). EtOAc (10 mL) was then added, and the phases separated. The aqueous layer was extracted with EtOAc (5 × 10 mL), and the combined organic layers were washed with brine (10 mL), dried over anhydrous Na_2_SO_4_, filtered, and concentrated in vacuo. The resulting crude material was immobilized on celite and purified by gradient flash chromatography (C18 silica, MeCN/H_2_O, 1:9 to 2:3), resulting in 58 mg (0.137 mmol, 60%) of the desired product as a yellow solid, mp. 144.1–145.4 °C, TLC (MeCN/H_2_O 2:1) *R_f_* = 0.19; ${\left[\alpha \right]}_{D}^{20}=-434$ (c 0.50, EtOH), HPLC purity > 98%, t_R_ = 10.69 min; ^1^H NMR (600 MHz, DMSO-*d_6_*) δ: 9.84 (s, 1H), 8.23 (s, 1H), 8.15 (d, *J* = 7.7 Hz, 1H), 7.95 (s, 1H), 7.59–7.47 (m, 4H), 7.44–7.32 (m, 2H), 6.96–6.85 (m, 2H), 5.44 (*p*, *J* = 7.1 Hz, 1H), 1.54 (d, *J* = 7.0 Hz, 3H); ^13^C NMR (151 MHz, DMSO-*d_6_*) δ: 164.5, 158.1, 155.4, 153.3, 144.3, 139.0, 131.2 (2C), 128.3 (2C), 127.2 (2C), 124.2, 119.6, 117.6, 116.1 (2C), 113.1, 48.7, 40.1, 39.9, 22.4; IR (neat, cm^-1^): 3321 (O-H), 2973 (Ar C-H), 1586 (N-H), 1495 (Ar C-H), 825 (Ar C-H); HRMS (TOF ES+, *m/z*): calcd. for C_20_H_17_BrN_3_OS [M + H]^+^: 426.0276, found: 426.0283.

### 3.5. Bioassays

#### 3.5.1. MIC Measurements 

The MIC of compound **1–21** towards *Escherichia coli* (MG1655) and *Staphylococcus aureus* (ATCC29213) was determined following the standards that were recommended by the Clinical and Laboratory Standards Institute (CLSI) for the broth microdilution method [40]. Briefly, the bacterial suspensions were adjusted to 0.5 McFarland standard (~1 × 10^8^ colony forming units (CFU)/mL) and diluted 1:200 in Cation-Adjusted Mueller-Hinton Broth (CAMHB, 22.5 mg/mL Ca^2+^, 11 mg/mL Mg^2+^). The suspension was subsequently added to polypropylene microtiter plates (100 µL/well, ~5 × 10^4^ CFU/well) that were already prepared with different concentrations of the various halogenated fused pyrimidines (11 µL/well, two-fold serial dilutions). The plates were incubated at 37 °C overnight before inspection for visible growth and determination of the MIC values.

#### 3.5.2. *E. coli* and Human TMPK Assay

The inhibition of *Ec*TMPK was determined by Profoldin using the *E*. *coli* Thymidylate Kinase Assay Kit Plus (ProFoldin, Catalog No. TMK100KE, Hudson, MA, USA), where dTMP was phosphorylated by ATP that was catalyzed by TMPK. The assay was run in 96-well black plates at 36 °C. The formation of ADP was measured by adding a fluorescent dye and measuring the fluorescence emission at 535 nm after excitation at 485 nm. The readout was corrected for emission from the assay cocktail and DMSO. An ADP control assay was also performed to correct for possible interference with ADP detection. Single-point inhibitions were measured at 8.3 µM inhibitor concentrations. The IC_50_-values were determined from a similar assay, where 2-fold dilution series from 200 mM to 0.391 mM were used. The assay towards the human enzyme was run similarly but using the Human Thymidylate Kinase Assay Kit Plus-500 (Catalog Number: HTMK500KE, ProFoldin, Hudson, MA, USA).

#### 3.5.3. In Vitro EGFR (ErbB1) Inhibitory Potency 

The compounds were supplied in a 10 mM DMSO solution, and enzymatic EGFR (ErbB1) inhibition potency was determined by Invitrogen (ThermoFisher) using their Z’-LYTE^®^ assay technology. [49] The compounds were tested for their inhibitory activity at 100 nM in duplicates. The IC_50_ values that were reported for **1** and **5** are based on the average of 2 titration curves (20 data points), and were calculated from activity data with a four-parameter logistic model using SigmaPlot (Windows Version 12.0 from Systat Software, Inc., Palo Alto, CA, USA). Unless stated otherwise, the ATP concentration that was used was equal to apparent K_M_.

#### 3.5.4. Kinase Panel 

The compounds were supplied in a 10 mM DMSO solution, and enzymatic kinase inhibition potency was determined by ThermoFisher (Invitrogen, Waltham, MA, USA) using their Z’-LYTE^®^ assay technology [49], at 500 nM in duplicates. The ATP concentration that was used was equal to K_m_, except when this service was not provided, and other concentrations had to be used.

## 4. Conclusions

A total of 21 different fused pyrimidines were synthesized and investigated for their antibacterial activity towards *E. coli* and *S. aureus*. The SAR study identified two highly active pyrrolopyrimidines with low MIC values towards *S. aureus*, while none were effective against *E. coli*. Moreover, the SAR study showed that only a minor alteration in the structure affected the activity profoundly, which indicates that the compounds act on a specific intracellular target rather than on the cellular membrane. A hydroxyl group on the *meta*- or *para* position of the 6-aryl unit was found crucial for activity, and heavy halogens (bromo and iodo) in the 4-benzylamine group was strongly potency inducing. Interestingly, when the most potent derivatives were evaluated in combination with the antimicrobial peptide betatide, a four-fold decrease in the MIC value was obtained, a strategy which might be promising for avoiding resistance. The detailed mode of action is currently not known. However, the front runner compound was shown to be a moderately active inhibitor towards *E. coli* TMPK in enzymatic assays and this is also a possible target in *S. aureus*. No major interferences with human kinases were found.

## Data Availability

Data is contained in this article or Appendix A.

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
