# Peer review of "Halogenated Pyrrolopyrimidines with Low MIC on *Staphylococcus aureus* and Synergistic Effects with an Antimicrobial Peptide"

_antibiotics, 2022, doi:10.3390/antibiotics11080984_

Round 1

Reviewer 1 Report

The authors described the synthesis of pyrolopyrimidine derivatives and their bioactivities against bacterial and human enzymes. The authors also summarized the SAR of the compounds.

I have several comments and questions:

  1. Page 1, line 47: I’m not sure if ‘de novo’ is an appropriate word. De novo is usually used for designing something not previously existing in nature. Can authors elaborate on this or edit the sentences to reflect the meaning more accurately?

  2. Page 8, line 229: sequence similarity of 34% is actually pretty low, they could be seen as pretty different families already. Can authors perform folding prediction studies on the two TMPK proteins to evaluate their structural similarity and based on the results edit the reasoning of why compounds can also be active for S. aureus TMPK ? You can use RosettaFold, OpenFold, etc. Another way to argue on this is, is there highly conserved sites?

  3. Page 9, line 243: I assume you want to say “EGFR inhibition activity”?

  4. Is halogenation at meta sites really critical, considering compound 2 also seems to show activity improvement? Could it be just an occupancy effect? If the methyl group of compound 2 is changed to something bigger and bulky, do you still see an improvement of bioactivity?

Author Response

Page 1, line 47: I’m not sure if ‘de novo’ is an appropriate word. De novo is usually used for designing something not previously existing in nature. Can authors elaborate on this or edit the sentences to reflect the meaning more accurately?

Response: De novo is the phrase biochemist use for this process. We therefore keep this phrasing.

Page 8, line 229: sequence similarity of 34% is actually pretty low, they could be seen as pretty different families already. Can authors perform folding prediction studies on the two TMPK proteins to evaluate their structural similarity and based on the results edit the reasoning of why compounds can also be active for S. aureus TMPK ? You can use RosettaFold, OpenFold, etc. Another way to argue on this is, is there highly conserved sites?

Response: The text has been modified to: “Thus, even though the sequence similarity is not high (34% by Protein Blast), TMPK could be a target also in S. aureus as the folding (see Figure S41 in Supporting information) and important residues in the catalytic domains: LID, P-loop and the DRX motif are highly conserved.  Two references are inserted. The folding similarity is shown in Figure S41 in the supporting information

Page 9, line 243: I assume you want to say “EGFR inhibition activity”?

Response: the reviewer is correct, this has been changed.

Is halogenation at meta sites really critical, considering compound 2 also seems to show activity improvement? Could it be just an occupancy effect? If the methyl group of compound 2 is changed to something bigger and bulky, do you still see an improvement of bioactivity?

Response: The reviewer might be correct, but we have not verified this by experiments. The following sentence has been added to section 2.4: “Further studies on variation of the R1 group must be performed to verify if the increase in activity is caused by the halogens or if it is purely a size effect”.

Reviewer 2 Report

The manuscript entitled " Halogenated pyrrolopyrimidines with low MIC on Staphylococcus aureus and synergistic effects with an antimicrobial peptide " presents synthesis and antibacterial activity of pyrrolopyrimidines derivatives.

In order to improve the quality of the manuscript, I suggest some corrections:

1. According to European Committee for Antimicrobial Susceptibility Testing,

the MIC value should be reported in "mg/L" instead of "μg/mL". Authors should correct it.

2. IR-spectra should confirm the structure, therefore, it should be cited signals which prove the structure, each characteristic signal should be assigned to some structural feature. The presented IR data contain a number of unidentified signals.

3. Authors had to prepare many compounds for the final synthesis. For better clarity of the manuscript, Authors should move the "Synthesis of building blocks" (4-Chloro-7-((2-(trimethylsilyl)ethoxy)methyl)-7H-pyrrolo[2,3-d]pyrimidine and compounds 22-42) to Supplementary Materials.

Author Response

In order to improve the quality of the manuscript, I suggest some corrections:

According to European Committee for Antimicrobial Susceptibility Testing,

the MIC value should be reported in "mg/L" instead of "μg/mL". Authors should correct it.

Response: The MIC values have been changed to mg/L

IR-spectra should confirm the structure, therefore, it should be cited signals which prove the structure, each characteristic signal should be assigned to some structural feature. The presented IR data contain a number of unidentified signals.

Response: Key IR signals have been assigned.

Authors had to prepare many compounds for the final synthesis. For better clarity of the manuscript, Authors should move the "Synthesis of building blocks" (4-Chloro-7-((2-(trimethylsilyl)ethoxy)methyl)-7H-pyrrolo[2,3-d]pyrimidine and compounds 22-42) to Supplementary Materials.

Response: This has been done.

Reviewer 3 Report

The manuscript submitted to Antibiotics journal investigates structure-activity relationship and showed the increased activity and selectivity after the insertion of heavy halogens (bromine and iodine). The result revealed synergetic effect reached by the most active agents in combination with betatide. However, there are some gaps and shortcomings in the manuscript, which must be corrected before considering its publication. Detailed comments for consideration are provided below:

Abstract part: The authors should point out the limiting factor and current situation of the infectious bacterial treatments. So, the readers can be convinced and agreed that we need new agent or drug combination.  A brief describes of experimental part should be specified here to provide the readers some idea how the results were obtained.

Introduction part: Too short. The authors should provide some literature reviews about the effect of halogen insertion on antibacterial activity or antimicrobial peptides in general. For the last paragraph, the information and background about experimental design should be mentioned. What kind of information we will obtained from E. coli TMPK in enzymatic assays and human kinases panel experiment? These info should be declared at this section.

Results part: In my opinion, the study design should be moved to “experimental section”.  2.2 Synthesis should be “pyrrolopyrimidines synthesis”. One sentence summarizing about the list of compounds that have been synthesized in this manuscript should be put in the first paragraph of this section.

Experimental section: I do not think that the phrase “General Procedures” is appropriate here. More specific descriptive subtopics should be used to engage the readers and prepare them what kind of “experiment” you are about to do at this part/procedure. All the chemical structures in 4.3 section should be put in the one table instead of figure instead of insertion in each paragraph like this. For the section 4.4 Bioassays, the citation of methodology papers should be placed in each subtopic either they have been taken or modified from the available protocols.

Acknowledgements: this section should be moved to the very last part, before the “Conflicts of Interest” part.

Author Response

The manuscript submitted to Antibiotics journal investigates structure-activity relationship and showed the increased activity and selectivity after the insertion of heavy halogens (bromine and iodine). The result revealed synergetic effect reached by the most active agents in combination with betatide. However, there are some gaps and shortcomings in the manuscript, which must be corrected before considering its publication. Detailed comments for consideration are provided below:

Abstract part: The authors should point out the limiting factor and current situation of the infectious bacterial treatments. So, the readers can be convinced and agreed that we need new agent or drug combination.  A brief describes of experimental part should be specified here to provide the readers some idea how the results were obtained.

Response: The abstract has been modified. More information on why new antibiotics are needed is given and the MIC values are included.

Introduction part:

a)Too short. The authors should provide some literature reviews about the effect of halogen insertion on antibacterial activity or antimicrobial peptides in general.

Response: Our search with SciFinder have not identified any reviews on small molecule halogenated antibiotics, only case studies. One review of relevance is on halogenated peptides/dipeptides. A reference has been included. The introduction already contains information on antimicrobial peptides. We feel that a further discussion on the effect of AMP is outside the scope of the paper and disrupt the flow of the introduction.

b)For the last paragraph, the information and background about experimental design should be mentioned. What kind of information we will obtained from E. coli TMPK in enzymatic assays and human kinases panel experiment? These info should be declared at this section.

Response: The following has been added at the end of the paragraph: “In the search of antibacterial targets an enzymatic assay towards E. coli and human TMPK was performed. Additionally, the most active derivative was evaluated toward a panel of 50 human kinases to identify off-targets.”

Results part: In my opinion, the study design should be moved to “experimental section”. 

Response: we have not done any change since we feel that this section is a relevant introduction to the study.

2.2 Synthesis should be “pyrrolopyrimidines synthesis”. One sentence summarizing about the list of compounds that have been synthesized in this manuscript should be put in the first paragraph of this section.

Response: as the synthesis also deals with thienopyrimidines, therefore we have entitled the chapter: “Synthesis of fused pyrimidines”.

Experimental section: I do not think that the phrase “General Procedures” is appropriate here. More specific descriptive subtopics should be used to engage the readers and prepare them what kind of “experiment” you are about to do at this part/procedure.

Response: This section has been entitled: Synthetic protocols

All the chemical structures in 4.3 section should be put in the one table instead of figure instead of insertion in each paragraph like this.

Response: The chemical structures have been removed. We have never seen a table of structures in any experimental procedure, so this suggested change has not been adapted.

For the section 4.4 Bioassays, the citation of methodology papers should be placed in each subtopic either they have been taken or modified from the available protocols.

Response: references has been placed in heading.

Acknowledgements: this section should be moved to the very last part, before the “Conflicts of Interest” part.

Response: corrected